# IterefinE: Iterative KG Refinement Embeddings using Symbolic Knowledge

**Siddhant Arora**                                        siddhantarora1806@gmail.com
*Dept. of CSE, IIT Delhi*

**Srikanta Bedathur**                                       srikanta@cse.iitd.ac.in
*Dept. of CSE, IIT Delhi*

**Maya Ramanath**                                        ramanath@cse.iitd.ac.in
*Dept. of CSE, IIT Delhi*

**Deepak Sharma**                                        dsharma080@gmail.com[*]

## Abstract

Knowledge Graphs (KGs) extracted from text sources are often noisy and lead to poor performance in downstream application tasks such as KG-based question answering. While much of the recent activity is focused on addressing the sparsity of KGs by using embeddings for inferring new facts, the issue of cleaning up of noise in KGs through *KG refinement* task is not as actively studied. Most successful techniques for KG refinement make use of inference rules and reasoning over ontologies. Barring a few exceptions, embeddings do not make use of ontological information, and their performance in KG refinement task is not well understood. In this paper, we present a KG refinement framework called **IterefinE** which iteratively combines the two techniques – one which uses ontological information and inferences rules, *viz.,*PSL-KGI, and the KG embeddings such as ComplEx and ConvE which do not. As a result, IterefinE is able to exploit not only the ontological information to improve the quality of predictions, but also the power of KG embeddings which (implicitly) perform longer chains of reasoning. The IterefinE framework, operates in a co-training mode and results in *explicit type-supervised* embeddings of the refined KG from PSL-KGI which we call as **TypeE-X**. Our experiments over a range of KG benchmarks show that the embeddings that we produce are able to reject noisy facts from KG and at the same time infer higher quality new facts resulting in upto 9% improvement of overall weighted F1 score.

## 1. Introduction

Knowledge graphs (KGs) represent facts as a set of directed edges or triples ⟨s,r,o⟩ where r is the relation between entities s and o. A critical issue in large-scale KGs is the presence of noise from the automatic extraction methods used to populate them. For instance, NELL [Carlson et al., 2010] is known to contain various kinds of errors including: different names for the same entity (e.g., australia and austalia), incorrect relationships –both due to wrong relation label as well as incorrect linkage altogether– between entities (e.g., ⟨matt_flynn, athleteplayssport, baseball⟩ is *false* since Matt Flynn is an NFL player), incompatible entity types, and many more [Pujara et al., 2013]. It has also been observed that such noise can significantly degrade the performance of KG embeddings [Pujara et al., 2017].

The *KG refinement* task aims to reduce the noise in KG by not only predicting additional links (relations) and types for entities (i.e., performing *KG completion*), but also eliminating incorrect facts. Methods for noise reduction in KG include the use of association rule mining over the noisy KG to induce rules which can help in eliminating incorrect facts [Ma et al., 2014]; reconciling diverse evidence from multiple extractors [Dong et al., 2014]; the use of ontology reasoners [Nakashole et al., 2011] and many more. A detailed survey of approaches for KG refinement is available in [Paulheim, 2017]. On the other hand, neural and tensor-based embeddings have seen significant success in

---

*. Work done at IIT Delhi.

entity type and new fact predictions [Nickel et al., 2012, Trouillon et al., 2016, Dettmers et al., 2018]. It is worth noting that embeddings, with a few recent exceptions [Guo et al., 2016, Minervini et al., 2017, 2018, Fatemi et al., 2019], do not make use of rich taxonomic/ontological rules when available. Methods such as Probabilistic Soft Logic (PSL) and Markov Logic Network (MLN) have been adapted for the KG refinement problem. They can address both the completion as well as noise removal stages of the KG completion problem. They can also make use of ontological rules effectively, and specifically, the PSL-KGI implementation uses rules defined on schema-level features [Pujara et al., 2013].

## 1.1 Contributions

In this paper we investigate the combined use of ontologies and embeddings in the KG refinement task. Ontologies are among the best methods to eliminate noisy facts in KGs, while embeddings provide a means of *implicitly* reasoning over longer chains of facts. Specifically, we use Probabilistic Soft Logic (PSL) that can incorporate inference rules and ontologies, along with state-of-the-art KG embedding methods,*viz.,* ConvE [Dettmers et al., 2018] and ComplEx [Trouillon et al., 2016], which do not make use of any ontological rules.

The resulting framework called **IterefinE** is based on the observation that the mispredictions by the embeddings based methods are often due to the lack of type compatibility between the entities due to their type-agnostic nature [Xie et al., 2016, Jain et al., 2018]. Since PSL-KGI is able to predict entity types by making use of ontological information along with many candidate facts derived using its inference rules, **IterefinE** transfer these predictions from PSL-KGI to the embeddings. This results in embeddings with *explicit* type supervision, which we call as **TypeE-ComplEx** and **TypeE-ConvE**. Further, we feed the predictions back from TypeE-ComplEx (correspondingly, TypeE-ConvE) over the training set to the PSL-KGI, resulting in additional evidence for inference. This feedback cycle can be repeated for multiple iterations, although we have observed over various benchmark datasets that the performance stabilizes within 2 to 3 iterations. Our key findings reported in this paper are as follows:

(i) Explicit type supervision improves the weighted F1-score of embeddings by up to 9% over those which do not have type supervision.

(ii) Explicit type supervised models also outperform the implicit type supervised models [Jain et al., 2018]. The margin of improvement is large when the ontological information is sufficiently rich to begin with.

(iii) Rich ontological information is a critical ingredient for the performance of TypeE-ConvE and TypeE-ComplEx, particularly when we consider their ability to remove the noisy triples. We observed that on datasets like YAGO3-10 and FB15K-237, we improved F1 scores on noisy triples by 30% to 100%.

We note that, although we have experimented with ConvE and ComplEx, it easy to instantiate IterefinE to work with other embeddings, which we plan to explore in our future work.

## 2. Related Work

In this section, we describe how KG refinement is accomplished by methods based on inference rules and embeddings-based methods. There are other research directions for (partially) solving the KG refinement problem such as rule induction [Ma et al., 2014], classification with diverse extractors [Dong et al., 2014], crowdsourcing, etc., (see [Paulheim, 2017] for an overview). While these works have their own strengths and weaknesses, our focus in this paper is on the use of ontological rules (exemplified by PSL-KGI) and embeddings (we use ComplEx, ConvE and [Jain et al., 2018]). Rule induction methods are orthogonal to our work, and may augment or replace the

set of rules we use. Further, evidence from diverse extractors as in the case of [Dong et al., 2014] can be incorporated into the PSL-KGI framework in a straightforward manner (see details about confidence values of triples in the Background section).

## 2.1 KG Refinement with Ontological Rules

Methods based on Markov-Logic Networks or Probabilistic Soft Logic (PSL), model the KG refinement task as a constrained optimization problem that scores facts in the KG with the help of various symbolic (logical) rules. An important input to these formulations are the probabilistic sources of information such as the confidence scores obtained during extraction [Pujara et al., 2013, Jiang et al., 2012] from multiple sources.

Of these methods, PSL-KGI [Pujara et al., 2013, 2017] is shown not only to perform better with KG noise and sparsity, but also to be quite scalable. It uses the following sources of information in addition to the noisy input KG: confidence scores of extractions, a small seed set of manually labeled correct facts and type labels and ontology information and inference rules.

## 2.2 Refinement task with KG embeddings

KG embedding methods define a scoring function $f$ to score the plausibility of a triple[1] and learn embeddings in such a way as to maximise the plausibility of the triples that are already present in the KG [Nickel et al., 2011, Socher et al., 2013, Trouillon et al., 2016].

An important step in learning is the generation of negative samples since the existing triples are all labeled positive. The negative samples are typically generated by corrupting one or more components of the triple. With this dataset containing both positive and negative samples, training can be done for the refinement task with a negative log-likelihood loss function as follows [Trouillon et al., 2016].

$$L(G) = \sum_{(s,r,o,y)\in G} y \log f(s,r,o) + (1-y)\log\left(1 - f(s,r,o)\right) \tag{1}$$

where $(s,r,o)$ is the relation triple, $f$ is the scoring function, and $y$ denotes whether the triple is given positive label or negative. Similar to the setting for PSL-KGI, embedding-based methods can also be used to predict type labels of entities (the *typeOf* relation). We work with ComplEx [Trouillon et al., 2016] and ConvE [Dettmers et al., 2018] embeddings which have shown state of the art performance in many KG prediction tasks.

## 2.3 Type and Taxonomy Enhanced Embeddings

There are some recent efforts to incorporate type hierarchy information in KG embeddings –e.g., TKRL [Xie et al., 2016] and TransC [Lv et al., 2018]. Recently, SimplE$^+$ [Fatemi et al., 2019] was introduced, which includes taxonomic information –i.e., subtype and subproperty information– and the authors also show that state-of-the-art embeddings like ComplEx [Trouillon et al., 2016], SimplE [Kazemi and Poole, 2018], ConvE [Dettmers et al., 2018] cannot enforce subsumption.

Taking a different approach [Jain et al., 2018] propose extending standard KG embeddings *without explicit* type supervision by representing entities as a two-part vector with one part encoding only the type information while the other one is a traditional vector embedding of the entity (and corresponding change to the relation embeddings as well). Specifically it uses the following scoring function :

$$f(s,r,o) = \sigma(\mathbf{s_t} \cdot \mathbf{r_h}) * \mathbf{Y}(s,r,o) * \sigma(\mathbf{o_t} \cdot \mathbf{r_t}), \tag{2}$$

---

1. See [Wang et al., 2017] for a survey of embedding methods and the many forms the scoring function $f$ can take.

where $\mathbf{s_t}$ and $\mathbf{o_t}$ denote the embedding vectors for implicit type label of entities, and $\mathbf{r_h}$ and $\mathbf{r_t}$ denote the implicit type embeddings for domain and range of relation $r$. $\mathbf{Y}$ is the scoring function used by the underlying embeddings-based method – we experiment with ComplEx and ConvE.

These embeddings enforce type compatibilities during KG link prediction task, and [Jain et al., 2018] showed nearly 5-8 point improvements in MRR and type F1 scores. In our work, we build on this idea further by adding another layer of explicitly supervised type vector to learning entity/relation embeddings.

Note, however, that our focus in this paper is not on embeddings that enforce ontological constraints, but on improving the KG refinement by combining the strengths of KG embeddings with methods like PSL-KGI and MLNs which can work with arbitrary (first-order) constraints.

Recently, there has been some work in modeling structural as well as uncertainty information of relations in the embedding space. [Chen et al., 2019] uses Probabilistic Soft Logic to come up with plausibility scores for each fact which they train to match with the uncertainty score of seen relation triplets as well as minimize the plausibility score for relation triplets. However, they do not focus on the KG refinement task and they also do not investigate how existing Knowledge Graph Embedding methods can be used in conjunction with this approach to effectively embed Uncertain graphs. There has also been some research in using rule-based reasoning and KG embeddings together in an iterative manner in [Zhang et al., 2019]. They achieve improvements in the performance of link prediction tasks for sparse entities which cannot be effectively modelled by standard embedding methods. However, at each iteration, they are adding more rules to their database, which makes their approach less scalable than one proposed in the paper. Since we are continuously removing noise from Knowledge Graph, thus making the size of the resultant Knowledge Graph stable. Also, the feedback in their work was rules learned from embedding with a robust pruning strategy. In contrast, we passed feedback as relation triples along with their predicted score as additional context for the PSL-KGI model to generate high quality predictions. Finally, we test this feedback in Knowledge Graph refinement manner where we couple the task of removing noise as well as inferring new rules together in a coupled manner with both the tasks benefiting from each other.

## 3. Background

We use the PSL-KGI implementation generously provided by the authors[2], that takes as input:

(i) the triples extracted from multiple input sources and confidence values for these triples,

(ii) ontology information, such as sub-class (SUB) and sub-property (RSUB) information; the domain and range of relations (DOM, RNG); "same" entities (SAMEENT), entities and relations that are mutually exclusive (MUT and RMUT); and inverse relations (INV). We reproduce the list of information used in [Pujara et al., 2013] in tabular form in Table 1.

(iii) inference rules – specifically, there are 7 general constraints that were first introduced in the earlier work on Markov Logic Networks (MLN) based work [Jiang et al., 2012]. These rules are listed in Table 2.

Based on these PSL-KGI defines a PSL program that combines the ontological rules and constraints with atoms in the KG. The solution to the PSL program essentially provides most likely interpretation of the KG, defining a probability distribution over the KG. By appropriately selecting the threshold on the probability value, it is possible to reject noisy facts. It is also important to note that PSL-KGI also generates a number of candidate facts that are not originally in the KG by soft-inference over the ontology and inference rules. While the extraction confidence for a triple may be high, it is possible for PSL-KGI to output a low score for that triple because of the inference rules. As a result, PSL-KGI is able to determine correct type labels and expand the seed set iteratively.

---

2. https://github.com/linqs/psl-examples/tree/master/knowledge-graph-identification

| Ontological Information | Description |
| --- | --- |
| Domain (DOM) | Domain of a relation |
| Range (RNG) | Range of a relation |
| Same Entity (SAMEENT) | Helps perform entity resolution by specifying equivalence class of entities |
| MUT | Specifies that 2 entities are mutually exclusive in their type labels |
| Subclass (SUB) | Subsumption of labels |
| INV | Inversely related relations |
| RMUT | Mutually exclusive relations |
| SUBPROP (RSUB) | Subsumption of relations |

Table 1: Ontological Information used in PSL-KGI Implementation [Pujara et al., 2013]

| Class | Ontological Rule |
| --- | --- |
| Uncertain Extractions | $w_{CR-T} : CANDREL_T(E_1, E_2, R) \Rightarrow REL(E_1, E_2, R)$ 
 $w_{CL-T} : CANDLBL_T(E, L) \Rightarrow LBL(E, L)$ |
| Entity Resolution | $SAMEENT(E_1, E_2) \wedge LBL(E_1, L) \Rightarrow LBL(E_2, L)$ 
 $SAMEENT(E_1, E_2) \wedge REL(E_1, E, R) \Rightarrow REL(E_2, E, R)$ 
 $SAMEENT(E_1, E_2) \wedge REL(E, E_1, R) \Rightarrow REL(E, E_2, R)$ |
| INV | $INV(R, S) \wedge REL(E_1, E_2, R) \Rightarrow REL(E_2, E_1, S)$ |
| Selectional Preference | $DOM(R, L) \wedge REL(E_1, E_2, R) \Rightarrow LBL(E_1, L)$ 
 $RNG(R, L) \wedge REL(E_1, E_2, R) \Rightarrow LBL(E_2, L)$ |
| Subsumption | $SUB(L, P) \wedge LBL(E, L) \Rightarrow LBL(E, P)$ 
 $RSUB(R, S) \wedge REL(E_1, E_2, R) \Rightarrow REL(E_1, E_2, S)$ |
| Mutual Exclusion | $MUT(L_1, L_2) \wedge LBL(E, L_1) \Rightarrow \neg LBL(E, L_2)$ 
 $RMUT(R, S) \wedge REL(E1, E2, R) \Rightarrow \neg REL(E1, E2, S)$ |

Table 2: Ontological Inference Rules used by PSL-KGI [Pujara et al., 2013]

## 4. Combining PSL-KGI with KG embeddings

OVERVIEW

We now present a simple mechanism, partially based on the concept of co-training [Blum and Mitchell, 1998], to combine the strengths of PSL-KGI and KG embeddings. The mechanism consists of two stages, as shown in Figure 1. In the first stage, PSL-KGI is used to generate high-quality type predictions, and in the second stage, an enhanced KG embeddings method, which we term as TypeE-X (where **X** is an embeddings method such as ComplEx), takes as input, the type predictions and relation triples labeled *true* in the training set. At the end of the second stage, the embeddings generated are expected to be of higher quality. The feedback to PSL-KGI is completed by passing the predictions from the KG refinement of TypeE-X back to PSL-KGI which takes them, along with the original extraction scores, as additional context for predicting relation triples. Note that this process can be repeated iteratively, allowing the propagation of potentially more context at each iteration[3].

Our observations show that passing *all* newly predicted triples by TypeE-X back to PSL-KGI as feedback would make our approach nonscalable for multiple iterations. Therefore, we only add some of the top most positive and most negative relations so that the size of the KG remains stable without

---

3. For an algorithmic listing of IterefinE, please refer to Appendix A.3 [Arora et al., 2020]

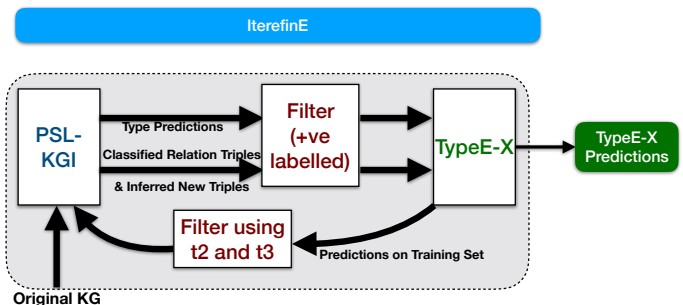

Figure 1: IterefinE: Combining PSL-KGI and embedding model X resulting in TypeE-X model.

sacrificing accuracy. In order to ensure that an optimal number of positive and negative triples are fed back to PSL-KGI, we calculate *separate* thresholds for each. First, the classifier threshold $t_1$ determines which triples are predicted as positive and which are negative. This threshold is determined by optimizing over a validation set. Second, we divide the set of triples, using $t_1$, into positive triples, denoted by $P_1$, and negative triples, denoted by $N_1$. Now, we choose two new thresholds $t_2$ and $t_3$:

$$t_2 = t_1 + \Phi_1 * mean(P_1)$$
$$t_3 = t_1 - \Phi_2 * mean(N_1)$$
(3)

where $mean(X)$ is the mean score of triples in set $X$, $\Phi_1$ and $\Phi_2$ are parameters that can be tuned. Then we add all relations with predicted probability greater than $t_2$, along with their inferred probabilities, as a form of positive feedback for our PSL-KGI model of the next iteration. Similarly, the negative feedback would consist of all relations with predicted probabilities less than $t_3$. We discuss the impact of these thresholds on the size of the KG and the prediction accuracy in Section 6.4.

SCORING FUNCTION FOR TYPEE-X

To incorporate the type inferences for entities generated by PSL-KGI in KG embeddings (the second stage), we modify the *typed model* [Jain et al., 2018] as follows:

Instead of just using the implicit type embeddings, we concatenate them with embeddings of explicit types transferred from PSL-KGI. Note that the implicit type embeddings are learned for each entity or relation, whereas the explicit type embeddings are the same for all entities with the same type label. The scoring function for extended typed model, TypeE-X, with an underlying embedding model **X** is

$$f(s,r,o) = \sigma((\mathbf{s_t}\|\mathbf{s_l}) \cdot (\mathbf{r_h}\|\mathbf{r_{dom}})) * \mathbf{Y}(s,r,o) * \sigma((\mathbf{o_t}\|\mathbf{o_l}) \cdot (\mathbf{r_t}\|\mathbf{r_{range}})),$$
(4)

where $\mathbf{s_l}$ denotes the explicit type label assigned to entity $s$, $\mathbf{r_{dom}}$ and $\mathbf{r_{range}}$ provide the explicit type labels for domain and range of a relation respectively. The type compatibility is enforced by *concatenating*, denoted $\|$, the two vectors and taking their dot product. In case an explicit type label for an entity is unknown, we use the *UNK* embedding as per the convention.

## 5. Preparing Datasets for Evaluating the KG Refinement Task

Before we present the details of the datasets used in our study, we first present the methodology followed to prepare them for use in the KG refinement task. As discussed earlier, apart from NELL,

none of the KG benchmarks contain noise labels, making them unsuitable for evaluating the KG refinement task. We prepare them as follows:

- We sample a random 25% of all facts (including the `typeOf` relations) and corrupt them by randomly changing their subject, relation label or object. Note that this was the same model followed in an earlier study [Pujara et al., 2017].

- We further refine the noise model by ensuring that *half* of the corrupted facts have entities that are type compatible to the relation of the fact. This makes it harder for detecting corrupted facts simply by using type compatibility checks.

To capture realistic KG refinement settings, we further add extraction scores generated by sampling them from two different normal distributions: $N(0.7, 0.2)$ for facts in the original KG and $N(0.3, 0.2)$ for added noisy facts [Pujara et al., 2013]. The SAMEENT facts between entities are generated by calculating the average of the two Jaccard similarity score over sets of relationships with these pair of entities as head and tail entity respectively – the average score acts as the confidence score of the fact. Finally, for all datasets, the test and validation sets are created by randomly partitioning the KG. Note that for all datasets the test set also includes the facts that were part of the original benchmark test collection.

### 5.1 Datasets

**NELL:** The NELL subset taken from its $165^{th}$ iteration [Carlson et al., 2010]) has been used for the KG refinement task [Pujara et al., 2013, Jiang et al., 2012]. It comes with a rich ontology from the NELL system, and contains multiple sources of information i.e., a single fact is present with multiple extraction scores. Since the original dataset does not have validation set, we split the test set into 2 equal halves preserving the same class balance, and use them as our validation and test split.

**YAGO3-10:** YAGO3-10 [Dettmers et al., 2018] is a subset of the YAGO3 [Suchanek et al., 2007] knowledge graph. It is often used for evaluating the KG completion task. We have augmented it with ontological facts and entity types derived from YAGO3. Since YAGO3 has a large number of types, we contract the type hierarchy to make it comparable to other datasets. We linked YAGO facts directly with the YAGO taxonomy by skipping the rdf:type entities at leaves of taxonomy (from YAGO simple types) and the first level of YAGO taxonomy. Then all facts upto length 3 in the hierarchy of taxonomy were included.

**FB15K-237:** FB15K-237 [Dettmers et al., 2018], another popular benchmark does not have ontological and type label information. Therefore, we use the type labels for entities from [Xie et al., 2016] which also provides the domain and range information for relations. The subclass information is populated by reconstructing the type hierarchy from type label facts. Mutually exclusive labels, relations and inverse relations are automatically created by mining the KG – e.g. we can find inverse relations by checking if all reverse edges exists in the KG for a relation.

**WN18RR:** WN18RR, similar to FB15K-237, does not contain ontological and type information. We used the synset information obtained from [Villmow, 2018], to assign type labels for entities. For example, for synset `hello.n.01`, the type is considered as `noun(n)`. Using an older ontology[4] we derived the rest of ontological information for the dataset.

Table 3 summarizes the size of different KG datasets we use in our evaluation. Table 4 shows the amount of ontological information for each dataset. NELL and FB15K-237 have reasonably rich ontological information compared to YAGO3-10 and WN18RR.

---

4. https://www.w3.org/2006/03/wn/wn20/

| Dataset | $|E|$ | $|R|$ | #triples in train / valid / test |
|---------|-------|-------|----------------------------------|
| NELL | 820K | 222 | 1.02M / 4K / 4K |
| FB15K-237 | 14K | 238 | 246K / 27K / 30K |
| YAGO3-10 | 123K | 38 | 1.13M / 10K / 10K |
| WN18RR | 40K | 12 | 116K / 6K / 6K |

Table 3: Number of entities, relation types and triples in each dataset.

| Dataset | DOM | RNG | SUB | RSUB | MUT | RMUT | INV | SAMEENT |
|---------|-----|-----|-----|------|-----|------|-----|---------|
| NELL | 418 | 418 | 288 | 461 | 17K | 48K | 418 | 8K |
| FB15K-237 | 237 | 237 | 44K | 0 | 147K | 53K | 44 | 20K |
| YAGO3-10 | 37 | 37 | 828 | 2 | 30 | 870 | 8 | 20K |
| WN18RR | 11 | 11 | 13 | 0 | 0 | 66 | 0 | 20K |

Table 4: Number of instances of each ontological component in datasets considered.

## 6. Experimental Evaluation

We evaluate the performance of TypeE-X models in the KG refinement task, and compare them with ComplEx [Trouillon et al., 2016] and ConvE [Dettmers et al., 2018], two state-of-the-art KG embeddings methods, and PSL-KGI. We also use ComplEx and ConvE as base embedding models for our TypeE-X method to get TypeE-ComplEx and TypeE-ConvE respectively. We use a single hyper-parameter threshold as the cutoff for classifying a test triple based on the prediction score [Pujara et al., 2013]. Our experiments were run on Intel(R) Xeon(R) x86-64 machine with 64 CPUs using 1 NVIDIA GTX 1080 Ti GPU. We observe the average running time with TypeE-ComplEx to be between 25–100 minutes and with TypeE-ConvE to be between 120–420 minutes per iteration. The increased time observed for TypeE-ConvE experiments is because of the fact that ConvE takes longer time to train than ComplEx[5]. The hyper-parameter is tuned on the validation set and used unchanged for the test set. We use $\phi_1 = 0.5$ and $\phi_2 = 0.75$ in Equation 3 as these hyperparameters were found to work across a variety of datasets.

**Evaluation Metric:** Our main evaluation metric is the *weighted F1* (wF1) measure. The reason for this is that in the KG refinement task, there is an imbalance in the two classes – noisy facts and correct facts[6]. Weighted F1 is defined as the individual class F1 score weighted by the number of instances per class in the test set.

$$wF1 = w_1 * F1(l_1) + w_0 * F1(l_0) \tag{5}$$

where $w_k$ is the fraction of samples with label $k$ ($k \in \{0, 1\}$ in our setting), $F1(l_k)$ is the $F1$ score computed only for class $k$.

### 6.1 Baselines

In addition to the baselines ComplEx, ConvE and PSL-KGI, we compare our method with two other ensemble methods, described below.

**ConvE + ComplEx:** In the first stage, instead of using PSL-KGI for predictions, we use ConvE. These predictions (along with the original KG) are used as input to the second stage which used ComplEx. Note that this baseline combines to *similar* methods.

---

5. Additional scalability experiments are reported in Appendix A.1 [Arora et al., 2020]
6. Noisy facts are much lower in number compared to correct facts.

| Method | NELL | YAGO3-10 | FB15K-237 | WN18RR |
|---|---|---|---|---|
| $\alpha$ - ComplEx | 1.0 | 0.4 | 0.7 | 0.3 |
| $\alpha$ - ConvE | 1.0 | 0.4 | 0.6 | 0.9 |

Table 5: Optimal $\alpha$ values obtained based on performance on validation set

| Method | NELL | | | YAGO3-10 | | | FB15K-237 | | | WN18RR | | |
|---|---|---|---|---|---|---|---|---|---|---|---|---|
| | +ve F1 | -ve F1 | wF1 | +ve F1 | -ve F1 | wF1 | +ve F1 | -ve F1 | wF1 | +ve F1 | -ve F1 | wF1 |
| ComplEx | 0.82 | 0.58 | 0.73 | 0.94 | 0.43 | 0.88 | 0.96 | 0.4 | 0.92 | 0.93 | 0.26 | 0.86 |
| ConvE | 0.74 | 0.55 | 0.67 | 0.94 | 0.37 | 0.87 | 0.95 | 0.37 | 0.90 | 0.93 | 0.07 | 0.84 |
| PSL-KGI | 0.85 | **0.68** | **0.79** | 0.91 | 0.39 | 0.85 | 0.92 | 0.39 | 0.88 | 0.91 | **0.37** | 0.85 |
| ConvE +ComplEx | 0.82 | 0.58 | 0.73 | 0.95 | 0.43 | 0.89 | 0.96 | 0.39 | 0.92 | 0.93 | 0.15 | 0.85 |
| $\alpha$ - ComplEx | 0.85 | 0.68 | 0.79 | 0.94 | 0.50 | 0.89 | 0.96 | 0.58 | 0.93 | 0.94 | 0.24 | 0.87 |
| $\alpha$ - ConvE | 0.85 | 0.68 | 0.79 | 0.94 | 0.41 | 0.88 | 0.95 | 0.47 | 0.92 | 0.92 | 0.34 | 0.85 |
| **TypeE-ComplEx** | **0.86** | **0.68** | **0.79** | **0.95** | **0.56** | **0.91** | **0.98** | **0.82** | **0.97** | 0.93 | 0.24 | 0.85 |
| **TypeE-ConvE** | **0.86** | 0.67 | **0.79** | **0.95** | 0.47 | 0.89 | **0.98** | 0.77 | 0.96 | **0.94** | 0.31 | **0.87** |

Table 6: Overall performance of all models in KG refinement task using the best wF1 measure obtained in first 6 iterations. +ve F1 indicate the F1 score for correct facts and -ve F1 indicate F1 score for noisy facts.

$\alpha$ **-model:** This baseline is a simple score combination of two different methods (in contrast to the two stages with iterations of our method). We use the setting introduced in R-GCN (Schlichtkrull et al. [2018]) to combine scores of KG embeddings and PSL-KGI methods using the equation given below:

$$f(h,r,t)_{\alpha-model} = \alpha * f(h,r,t)_{PSL-KGI} + (1-\alpha) * f(h,r,t)_{model} \qquad (6)$$

Here the hyperparameter $\alpha$ is chosen based on the validation set. The optimal $alpha$ value obtained are reported in table 5 and $model$ could be either ComplEx or ConvE.

### 6.2 Accuracy of TypeE-X

Our main results are shown in Table 6. We include separate F1 measures for the two classes as well as the weighted F1 measure. This helps us analyse how well each method performs in identifying the correct (+ve) and noisy facts (-ve). From the table, we observe that our proposed combined methods TypeE-X consistently outperform the KG embeddings methods as well as the baseline PSL-KGI. Note that PSL-KGI is a formidable baseline over NELL since it contains a rich ontology.

For the positive class (correct facts), our method performs slightly better than the second best competitor, while for the negative class (noisy facts), both our methods show substantial improvements for YAGO3-10 and FB15K-237 datasets, while performing on par with PSL-KGI for NELL. The only dataset on which our methods fail to beat the PSL-KGI baseline is WN18RR and this is because of its very limited ontology (please refer to Table 4). Further, for all datasets our TypeE-X methods have the best $wF1$ numbers. We have therefore validated our initial hypothesis that ontological information of high quality is tremendously helpful in improving the quality of embeddings.

**Comparison with Baseline Ensemble Models.** From Table 6, we see that TypeE-X models perform much better than ConvE +ComplEx. We hypothesize that this is because PSL-KGI and embeddings methods are complementary in nature. That is, PSL-KGI is better at removing noisy facts, while embeddings methods are better at inferring new facts. In contrast, when we combine

| Dataset | [Jain et al., 2018] | TypeE-ComplEx |
|---|---|---|
| NELL | 0.60 | **0.71** |
| YAGO3-10 | 0.88 | **0.92** |
| FB15K-237 | 0.93 | **0.97** |
| WN18RR | 0.85 | **0.85** |

Table 7: Weighted F1 scores on relation triples in the test set by [Jain et al., 2018] and TypeE-ComplEx.

ComplEx and ConvE, the resultant model cannot incorporate rich ontological information and, hence, cannot effectively remove noise from the KG. This intuition is confirmed by looking at low -ve F1 of these methods when compared to TypeE-X models in Table 6[7].

We also observe that $\alpha$-models perform better than the corresponding individual methods, but not better than our TypeE-X methods. This observation shows that our methodology of combining the two approaches in a pipeline fashion is more powerful than a simple weighted combination of these methods. The reason is that, in our method, each of the individual methods benefits from the strength of the other method since the results of one are used as input for the other. As a result, both these methods gain from each other's performance. Further, we list the computed alpha values that showed the best validation performance in Table 5. We observe that the alpha values are mostly tilted towards the better-performing model.

**Comparison with unsupervised type inference.** In Table 7 we compare the performance of TypeE-ComplEx which has *explicit* type supervision with the *unsupervised* type-compatible embeddings-based method proposed by Jain et al. [Jain et al., 2018]. As these results indicate, while explicitly ensuring type compatibility helps to improve performance, adding type inferences from PSL-KGI to TypeE-ComplEx significantly improves the relation scores, improving weighted F1 up to 18% (over NELL).

**Anecdotes.** Looking at the example predictions by both TypeE-ComplEx and ComplEx on YAGO3-10, we observed that TypeE-ComplEx is able to correctly identify simple noisy facts like ⟨Leinster_Rugby, hasGender, Republic_of_Ireland⟩ , where there is a clear type incompatibility, which ComplEx is unable to identify. Further TypeE-ComplEx is able to identify noisy facts such as ⟨Richard_Appleby, playsFor, Sporting_Kansas_City⟩, with type compatible entities, by finding the reasoning context that connect Richard_Appleby with football teams of UK and not US.

### 6.3 Analysis of Feedback Iterations

We have already shown in Table 6 that our TypeE-X methods output higher quality predictions compared to the other baselines. In this section, we analyse the conditions under which multiple iterations can improve the quality of predictions.

Figure 2 shows how the $wF1$ values of our TypeE-X methods change over six feedback iterations. Recall from Figure 1 that each iteration involves adding high quality tuples from PSL-KGI inferences to TypeE-X and feeding back high quality tuples from TypeE-X predictions back to PSL-KGI.

The main observation we make in Figure 2 is that the accuracy of predictions on datasets with a rich and good quality ontology (NELL, FB15K-237 and YAGO3-10) do not do not vary much. In fact, for NELL and YAGO3-10 the accuracy actually increases in multiple iterations (best accuracy for NELL is in the $6^{th}$ iteration, and for YAGO3-10 it is in the $3^{rd}$), while for FB15K-237, there is only a small decrease over the first and last iterations.

---

7. For more observations regarding noise removal, please refer to Appendix A.2 [Arora et al., 2020]

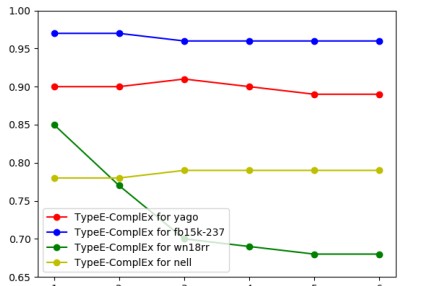 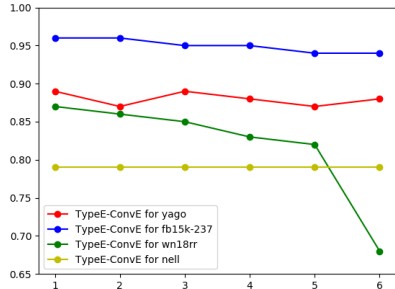

Figure 2: Graph showing the wF1 scores (y-axis) obtained at different feedback iterations (x-axis).

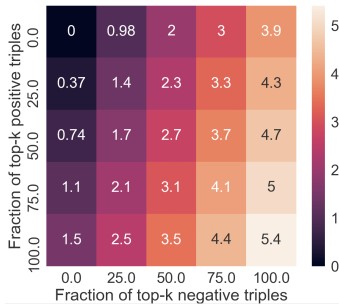 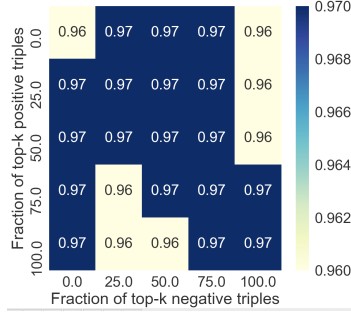

Figure 3: Variation of size (left) and variation in wF1 (right) with percentage of top positive and negative triples for TypeE-ComplEx after the first feedback iteration on FB15K-237.

In contrast, for the WN18RR dataset, the accuracy degrades quite rapidly after the first iteration. The reason is that this dataset does not have even a moderate number of ontological rules that are of high quality[8]. This results in lower quality inference from PSL-KGI which feeds into the TypeE-X method. This results in lower quality predictions from TypeE-X, which is then fed back into PSL-KGI. Thus a cascading effect of low quality predictions from each method results in a rapid drop in prediction quality.

### 6.4 Impact of Hyper-parameters ($t_1$ and $t_2$)

The threshold parameters $t_1$ and $t_2$ determine how many positive and negative triples are fed back to PSL-KGI from TypeE-X. The number of such feedback triples has an impact on both, the size of the KG as well as the accuracy of predictions (because PSL-KGI now performs inference using the new triples that have been fed back). Figure 3 shows, for FB15K-237, two heatmaps which quantify the impact of $t_1$ and $t_2$. In the left heatmap, the impact of adding the top-$k$ percent of positive and negative tuples on the *size* of the KG is shown[9] and in the right heatmap, the impact on the *accuracy* is shown. We observe that by adding very few positive and negative tuples, with slightly more positive tuples than negative tuples as feedback is sufficient to obtain the best accuracy, while ensuring that the KG size does not explode.

---

8. Recall from Section 5 that the ontology rules were obtained from [Villmow, 2018] and an older ontology.

9. The size is normalized: $\frac{(newsize - originalsize)}{(originalsize)}$

| Method | NELL | | | FB15K-237 | | |
|---|---|---|---|---|---|---|
| | +ve F1 | -ve F1 | wf1 | +ve F1 | -ve F1 | wF1 |
| All rules | **0.86** | **0.68** | **0.79** | **0.98** | 0.80 | **0.97** |
| No rules | 0.82 | 0.58 | 0.73 | 0.96 | 0.4 | 0.92 |
| w/o DOM | 0.85 (-0.01) | 0.65 (-0.03) | 0.78 (-0.01) | **0.98** (0.00) | 0.76 (-0.04) | 0.96 (-0.01) |
| w/o SAMEENT | 0.85 (-0.01) | 0.67 (-0.01) | **0.79** (0.00) | **0.98** (0.00) | 0.80 (0.00) | **0.97** (0.00) |
| w/o MUT | 0.85 (-0.01) | **0.68** (0.00) | **0.79** (0.00) | **0.98** (0.00) | 0.80 (0.00) | **0.97** (0.00) |
| w/o RNG | 0.82 (-0.04) | 0.65 (-0.03) | 0.76 (-0.03) | 0.97 (-0.01) | 0.72 (-0.08) | 0.95 (-0.02) |
| w/o SUB | 0.84 (-0.02) | 0.63 (-0.05) | 0.77 (-0.02) | **0.98** (0.00) | **0.81** (0.01) | **0.97** (0.00) |
| w/o RMUT | **0.86** (0.00) | 0.67 (-0.01) | **0.79** (0.00) | **0.98** (0.00) | 0.80 (0.00) | **0.97** (0.00) |
| w/o INV | 0.85 (-0.01) | 0.66 (-0.02) | 0.78 (-0.01) | **0.98** (0.00) | **0.81** (0.01) | **0.97** (0.00) |
| w/o RSUB | **0.86** (0.00) | 0.67 (-0.01) | **0.79** (0.00) | **0.98** (0.00) | 0.80 (0.00) | **0.97** (0.00) |
| ONLY DOM+RNG | 0.84 (-0.02) | 0.65 (-0.03) | 0.77 (-0.02) | **0.98** (0.00) | 0.80 (0.00) | **0.97** (0.00) |
| ONLY DOM | 0.84 (-0.02) | 0.64 (-0.04) | 0.77 (-0.02) | **0.98** (0.00) | 0.73 (-0.07) | 0.96 (-0.01) |
| ONLY RNG | 0.83 (-0.03) | 0.63 (-0.05) | 0.76 (-0.03) | **0.98** (0.00) | 0.76 (-0.04) | 0.96 (-0.01) |
| ONLY SAMEENT | 0.83 (-0.03) | 0.63 (-0.05) | 0.76 (-0.03) | **0.98** (0.00) | 0.73 (-0.07) | 0.96 (-0.01) |
| ONLY MUT | 0.83 (-0.03) | 0.63 (-0.05) | 0.76 (-0.03) | **0.98** (0.00) | 0.73 (-0.07) | 0.96 (-0.01) |
| ONLY SUB | 0.82 (-0.04) | 0.60 (-0.08) | 0.74 (-0.05) | **0.98** (0.00) | 0.74 (-0.06) | 0.96 (-0.01) |
| ONLY RMUT | 0.83 (-0.03) | 0.62 (-0.06) | 0.76 (-0.03) | **0.98** (0.00) | 0.76 (-0.04) | 0.96 (-0.01) |
| ONLY INV | 0.84 (-0.02) | 0.63 (-0.05) | 0.76 (-0.03) | **0.98** (0.00) | 0.73 (-0.07) | 0.96 (-0.01) |
| ONLY RSUB | 0.83 (-0.03) | 0.62 (-0.06) | 0.76 (-0.03) | - | - | - |

Table 8: Performance with/without different ontology components in KG refinement for TypeE-ComplEx. Results are for FB15K-237 at 2nd epoch and NELL at 3rd epoch.

### 6.5 Ablation Study

We performed an ablation study to determine what kind of ontology rules were most useful in increasing prediction accuracy. The results for two datasets, NELL and FB15K-237 are shown in Figure 8. From the table, we observe that it is the Subclass, Domain and Range rules that are the most important. Clearly these rules are most useful in correctly predicting types, which in turn are crucial for the accuracy of the TypeE-X methods.

Further, as these results show, none of the individual ontological components alone show performance comparable to using all the components (and thus all the rules) in the PSL-KGI phase of IterefinE. Although positive class performance over FB15K-237 remains unchanged when using *any* one ontological component, the performance over negative classes deteriorates significantly over using all the components. Thus, we argue that our proposal of using as much ontological information available in a KG is consistently superior for the KG refinement task.

## 7. Conclusion and Future work

We considered the KG refinement task and explored the crucial role played by ontology and inference rules on the performance of probabilistic rule based methods like PSL-KGI [Pujara et al., 2013]. We present an embedding framework called IterefinE that combines the PSL-KGI and type-supervised KG embeddings through iterative feedback. As a result IterefinE shows an overall increase in KG refinement performance over all datasets considered.

For future work, we plan to study approaches to train the entire pipeline end to end, thus increasing efficacy of our approach. Further we also plan to incorporate the KG data augmentation techniques to model richer context while training embeddings.

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
