# OpenReview forum: "IterefinE: Iterative KG Refinement Embeddings using Symbolic Knowledge"
_AKBC.ws/2020/Conference — AKBC 2020_

### Official Review · AnonReviewer2 · 2020-03-27
**More comparisons and insights about the results are desired**

**Rating:** 6
**Confidence:** 3

**Review:**

This paper tackles the task of knowledge-base refinement. The described method uses an approach based on co-training using a combination of two models. The first model PSL-KGI and the second model is a knowledge graph embeddings model (ComplEx or ConvE). The experiments are conducted on four datasets based on NELL, YAGO, FB15K, and WN18.

The idea of combining 2 conceptually different approaches like PSL-KGI and graph-embeddings work well, however, it is not surprising that it works better than a single model. It is observed in many works (this does not need a citation) that if you combine the prediction of multiple models into an ensemble model, it would work better than a single model. This is especially true if the models have a different nature. Additionally, a co-training setup, similar to the one presented here, would expectedly boost the performance further.
In that case, comparing the combined system to a single PSL-KGI or single KGE model is not enough. In order to claim some supreme performance of the method, it should be compared to similar methods that combine multiple models and ensembles of the same model types.

Some additional experiments are performed which could be of interest to the community with some further analysis and insights.

The analysis of the number of feedback iterations is interesting in order to know the limits of this type of refinement. It is very hard to see the difference from Figure 6 but for some of the datasets, ex. NELL it seems 6 steps do not seem to be enough to see the peak. Also, it is not clear if the difference in the performance is significant for most datasets. More insights into why more steps work or do not work are needed.

The ablation of types in Table 5  might be interesting but it needs further discussions what does it mean for a KG and each ontology that for example, RNG is the most important type. How does that help us to know more about them? And how is that related to the number of instances in each type (displayed in Table 2)?


Some minor comments:
- “It is worth noting that embeddings, with a few recent exceptions [Fatemi et al., 2019], do not make use of any form of taxonomic/ontological rules.” - there is work that uses taxonomy/rules with some examples being KALE (Guo et al 2016), ASR-X (Minervini 2017), NTP (Minervini et al 2020).
- In Table 5, please add the difference of the ablated results, to the overall (All) for better reading.

---

> ### Author Response · Authors · 2020-04-08
> **On why simple ensemble may not be sufficient, and other clarifications.**
>
> We thank all the reviewers for their valuable comments on our submission. We have revised our submission taking into consideration some of the review comments (revised content is specifically marked in blue for easy identification).
> 1. "In order to claim some supreme performance of the method, it should be compared to similar methods that combine multiple models and ensembles of the same model types. "
>      While it seems obvious (in hindsight), we too were surprised that no one has explored how to combine different models, in particular for combining ontological information with KG embeddings. Our experiments have shown that this is not only a natural framework with robust performance but also scalable, apart from being flexible to work with any KG embedding technique.
> Though we do not expand further in this paper, we experimented with two different kinds of combinations.
>     a) Combining ConvE + CompleX
>     b) Combining PSL-KGI and a KG embedding with a linear mixture [Schlichtkrull et al., 2017]: We constructed an ensemble model linearly mixing the scores of PSL-KGI and ComplEx to get $\alpha * score_{ComplEx} + (1-\alpha) * score_{PSL-KGI}$, with mixture parameter $\alpha$ optimized on the validation set (similarly for ConvE as well). We term the resulting model as $\alpha$-ComplEx and $\alpha$-ConvE respectively. Our results below show that IterefinE (TypeE-ComplEx and TypeE-ConvE) performs better compared to these baselines as well in both noise removal as well as overall.
>
> A brief tabulation of only the -ve F1 (performance of noise removal) and wF1 score is shared below (we show only FB15k-237 due to space):
>                                 FB15K-237
>                                -veF1  wF1
> ConvE+ComplEx   0.39   0.92
> α-ComplEx             0.58   0.93
> α-ConvE                  0.47   0.92
> TypeE-ComplEx     0.82   0.97
> TypeE-ConvE          0.77   0.96
> Our method of stage-wise combination of PSL and KG embeddings naturally combines the complementary strengths of these methods (PSL is better at removing noisy facts and KG embeddings are stronger at inferring new, and sometimes longer-range, facts) and performs at least as well as the second-best baseline and better than the baseline for some datasets (like FB15k-237 and YAGO).
>
> 2. "It is very hard to see the difference from Figure 6 but for some of the datasets, ex. NELL it seems 6 steps do not seem to be enough to see the peak. "
> We have tried to reason about when more steps work or do not work by trying to experiment with different datasets having different quality of ontological information. We observed that iterating more than once tends to improve the results (esp. on KGs with more ontological information).
>
> 3. "The ablation of types in Table 5 might be interesting but it needs further discussions what does it mean for a KG and each ontology that for example, RNG is the most important type. How does that help us to know more about them? And how is that related to the number of instances in each type (displayed in Table 2)?"
>
>    We disagree that DOM/RNG/SUBCLASS are the only important information In this revision, we have added another table, Table 8, which shows the performance using only one of the ontological component we have considered (along with DOM + RNG combined). As we can see from these results, using the only RNG is in fact no different than using only RMUT (on FB15K-237), or using any one of SAMEENT or MUT (on NELL). Thus, we argue that using all constraints from a rich ontology is important. We have added this discussion in the Appendix in the revised version.
>
> Now to look into how the number of instances of the type is correlated to performance, we performed the following experiment on FB15K-237. We retained ALL rules except for the RNG rules. We then gradually added RNG rules to see how increasing the number of ontological constraints would impact the performance. We tabulate these results below. Taken in conjunction with Table 8, this clearly shows that a combination of all possible rules gives us the best result – particularly to remove noisy rules (-ve F1).
>
> % of RNG rules     -ve F1     +ve F1       wF1
> 0%                 0.72        0.97       0.95
> 25%                0.74        0.98       0.96
> 50%                0.75        0.98       0.96
> 75%                0.77        0.98       0.97
> 100%               0.80        0.98       0.97
>
> 4 "there is work that uses taxonomy/rules with some examples being KALE (Guo et al 2016), ASR-X (Minervini 2017), NTP (Minervini et al 2020)."
>   Thanks for pointing out these references. We had previously experimented with NTP for the refinement task, but it failed to scale for large KG. ASR-X and KALE do not make a full set of ontological components as we do (through PSL-KGI) in IterifinE. We have revised the statement in the revised version by adding the suggested references.

---

> > ### Comment · AnonReviewer2 · 2020-04-29
> > **Thanks for the updates**
> >
> > Thanks to the authors about addressing some of my concerns. I think that with some further improvements the paper could be accepted.
> >
> > 1. On the reply of point 1:
> > The experiments with Conve+CompleX and the alpha ensembles should be reported in Table 3, including the actual alpha values (at least in the appendix). As I commented before - results for a combination of methods that improve over the single methods' runs is not a surprise.
> >
> > 2. It is not easy to see the difference between the number of iterations for some of the datasets. Plot the results in the range of the min iteration value - margin (ex. 0.7 instead of 0.5) to max iteration value because it is not clear what the difference is. It would be nice to also display the first iteration and the best iteration values on the plot.
> >
> > 3. It is nice that you added Table 8. However, in order to see the improvement, the results for every single type of rules should be compared to the baseline - ComplEx, not the experiments with all rules.

---

### Official Review · AnonReviewer1 · 2020-03-30
**Interesting augmentation of KG embeddings**

**Rating:** 6
**Confidence:** 4

**Review:**

Summary:
The authors propose a new method for identifying noisy triples in knowledge graphs that combines ontological information, using Probabilistic Soft Logic, with an embedding method, such as ConvE or ComplEx. They show that combining these two approaches improves performance on four different datasets with varying amounts of ontological information provided.

Clarity:
The paper is generally clear and well-written. I think it would be helpful to give a little more detail on how psl-kgi works. For example, it's not entirely clear to me how it outputs the type information.

Originality & Significance:
As mentioned in the paper, implicit type embeddings have been incorporated into embedding methods, but more extensive ontological information has not been used in this way. They also show that doing so results in improved performance over competitive baselines for this task.

Pros:
 - Novel use of ontological features with more recent embedding approaches for KG refinement
 - Performance improvement over competitive baselines

Cons:
 - The analysis feels a bit lacking. See comments below for more thoughts here.

Comments:
 - It doesn't seem like the iterative aspect of the model actually helps? From figure 2, it only appears to hurt performance
   on some datasets, and from figure 3, the change in accuracy appears to be minimal, and not obviously more than just
   noise.
 - I would be curious to see how much using the full ontology improves things vs just using explicit type information (DOM,
   RAN, and type labels for entities). Also, if most of the benefit comes from the type information, it might be interesting to
   see how much psl-kgi is actually adding, or whether you can just apply the type labels directly (for the datasets that you
   have them anyways).
 - While perhaps not in the scope of this paper, it would also be interesting to see how incorporating the ontological
   information affected other KG tasks, like link prediction.

---

> ### Author Response · Authors · 2020-04-08
> **Updated analysis, clarifications on the use of iterations, and link prediction results added.**
>
> We thank all the reviewers for their valuable comments on our submission. We have revised our submission taking into consideration some of the review comments (revised content is specifically marked in blue for easy identification).
>
> 1. "- The analysis feels a bit lacking. See comments below for more thoughts here.
> Comments:
> - It doesn't seem like the iterative aspect of the model actually helps? From figure 2, it only appears to hurt performance on some datasets, and from figure 3, the change in accuracy appears to be minimal, and not obviously more than just noise. "
>
>     We agree that adding too many iterations is not always beneficial across all datasets. However, it is important to note that for certain datasets that have good ontology (viz., NELL and Yago3-10) the best performance of IterefinE is after completing at least 3 iterations (Figure 2). Further, considering only the negative facts (Table 3), IterefinE requires up to 3 iterations to achieve nearly 30% improvement over the second-best baseline for YAGO3-10 and over 100% for FB15K-237.  Referring to Figure 5, noise removal in IterefinE improves only at 3rd iteration for NELL. For KGs such as WN18RR with poor ontology, the iterations are indeed not helpful. Therefore, the iterative framework we have introduced is expected to be beneficial for refinement tasks over KGs with ontologies.
>
> 2. "- I would be curious to see how much using the full ontology improves things vs just using explicit type information (DOM, RAN, and type labels for entities).  "
>
>     In the revised submission, we added another table (Table 6, Appendix A.4) where we show that the use of individual type information (DOM/RNG) or other ontological components perform poorer than the use of full ontology. This is particularly true when we consider the performance over the negative classes (i.e., the ability to remove noise). These results, in combination with those presented in Table 5, makes a strong case for using all constraints from a rich ontology.
>
>  Our experiment with the results of (Jain et al., 2018) shown in Table 4, was specifically to answer the question of whether just adding type information is sufficiently powerful enough to reduce noise as well. [Jain et al., 2018] learn implicitly type information when type information is not available explicitly in the KG, and use that to boost the performance of ComplEx.  On datasets that contain significant ontological information (all except WN18RR), our model is found to be superior.
>
> 3. "- While perhaps not in the scope of this paper, it would also be interesting to see how incorporating the ontological information affected other KG tasks, like link prediction. "
>
>    We were intrigued by the reviewer’s question about the performance of link prediction. Our focus in this work was on refining noisy knowledge graphs. Thus we felt that simply comparing the link prediction performance using IterefinE embeddings over SOTA models on FB15K-237/WN18RR/Yago3-10 will be unfair. Therefore, we designed two link prediction tasks:
>      i)  First, we compare the link prediction performance of IterefinE, CompleX and ConvE embeddings trained on noisy KG (same noisy KG that we run our experiments on). Results obtained over noisy FB15k-237 are shown in the table below. It is interesting to see that ConvE which is a strong baseline for link prediction on KGs without noise, performs the poorest over noisy KGs (it is an interesting finding which seems to require further investigation altogether). IterefinE (with CompleX embeddings) offers nearly 30% improvement in MRR over ComplEx and nearly 300% improvement over ConvE.
> Thus, we believe that IterefinE is attractive for link prediction task over noisy KG.
>
>                                                   MRR       Hits @ 10     Hits @ 1
> ConvE                                      0.0447     10.82%        1.34%
> ComplEx                                 0.1322     26.75%        7.16%
> IterefinE (TypeE-ComplEx    0.1691     28.92%        11.64%
>
>      ii) Second, we also wanted to explore how good IterefinE embeddings are for link prediction on KGs without any noise. We worked with FB15k-237 again, but this time on the original KG. Results are below:
>
>                                                                MRR    Hits @ 10    Hits @ 1
> ComplEx (Dettmers et al., 2018)       0.247   42.8%        15.8%
> IterefinE TypeE-ComplEx                  0.2532  43.55%       16.95%
>
> Here too IterefinE is visibly better -- although not as much as in the case of noisy KG--  than the previously reported numbers for ComplEx (Dettmers et al., 2018). Thus, even in the case of KGs without noise, IterefinE embeddings may potentially improve over SOTA with further tuning. We thank the reviewer for suggesting this task.

---

### Official Review · AnonReviewer4 · 2020-04-01
**Clear goal, good results, but needs more detail**

**Rating:** 6
**Confidence:** 1

**Review:**

The paper addresses the KG refinement task, aiming to improve KGs that were built via imperfect automated processes, either by removing incorrect relationships or by adding missing links. The key insight is to augment KG embeddings not only with implicit type information, but also with explicit types produced by an ontology-based system. The resulting algorithm, TypeE-X, leverages the benefits of structured (often human-crafted) information and the versatility of continuous embeddings.

The model proceeds in two stages. First, the PSL-KGI component (which takes in KG triples, ontology information and inference rules) produces type information for entities and relations. In the second stage, these are passed to the TypeE-X module, which appends this explicit type information to a) implicit type embeddings, and b) general-purpose KG embeddings. These two steps can optionally be repeated for multiple iterations, in a loop.

While the high-level picture is clear, there are a few details about the information flow and implementation that are harder to figure out:
- At the end of section 3, the authors write "It is also important to note that PSL-KGI also generates a number
of candidate facts that are not originally in the KG by soft-inference over the ontology and inference rules". It is not obvious in Figure 1 when this happens.
- How is this model parameterized? What exactly is trainable? In the conclusions sections, the authors write "we will look in ways to combine such methods at the training level", which raises the previous question again.
- How are the types produced by PSL-KGI converted to continuous representations? Is it a simple dictionary lookup?

The authors validate their models on four datasets, with ontologies of various sizes. They compare against multiple baselines, including PSL-KGI alone, generic KG embeddings alone, and generic KG embeddings + implicit type embeddings, showing their work outscores previous work.

One observation is that the datasets are "prepared" somewhat artificially (noise is programmatically inserted in the KGs, and the model is expected to detect these alterations), and it's not entirely clear how well this added noise correlates with the noise encountered in real-world KGs. It would be interesting to provide results on a downstream task (e.g. KG-based question answering) with and without KG refinement, to get an understanding of how much this step helps. However, in authors' defense, they are following the same procedure as previous work, and do make an effort to ensure the de-noising task is reasonably hard (e.g. "half of the corrupted facts have entities that are type compatible to the relation of the fact")

The ablation studies are insightful -- they look into how the number of loop iterations affect performance on various datasets, the impact of threshold hyper-parameters, and the impact of various ontological rules.

Overall, I think the paper is well written. It has a clear goal and convincing evidence to achieve it. However, I would have liked to see a clearer explanation of the algorithm and more implementation details.

---

> ### Author Response · Authors · 2020-04-08
> **Clarified the algorithm and information flow.**
>
> thank all the reviewers for their valuable comments on our submission. We have revised our submission taking into consideration some of the review comments (revised content is specifically marked in blue for easy identification).
>
> 1. "- At the end of section 3, the authors write "It is also important to note that PSL-KGI also generates a number of candidate facts that are not originally in the KG by soft-inference over the ontology and inference rules". It is not obvious in Figure 1 when this happens. "
>
>   Please note that PSL-KGI is used to do both: infer new facts, remove noisy facts. So, whenever, PSL-KGI runs, we do generate new facts. We have made changes in Figure 1 to show we are inferring new triples.
>
> 2. "- How is this model parameterized? What exactly is trainable? In the conclusions sections, the authors write "we will look in ways to combine such methods at the training level", which raises the previous question again. "
>
>   The parameters are learned at each stage, optimizing the objective of the stage. First, we train PSL-KGI and learn its parameters, use it to refine the KG. Note that this refinement step of PSL-KGI is essentially constructing a most likely interpretation of the KG or soft-truth assignments to entities, labels, and relations that comprise the KG, optimizing the performance over validation. This refined KG is fully materialized and is given as input to the model in the next stage (TypeE-ComplEx or TypeE-ConvE) and the parameters of this model along with embeddings for each explicit type as well as the implicit type/domain/range embeddings (Jain et al. 2018) are then learned. The KG is again filtered and fed as input to PSL-KGI, thus the process continues. We have added an algorithm 1 in the Appendix A.5 to make this process clearer.
>
> 3. "- How are the types produced by PSL-KGI converted to continuous representations? Is it a simple dictionary lookup?"
>   Please see the explanation to (2) above. The output of PSL-KGI is fully materialized as a KG with appropriate type assignments to all entities.
>
> 4. "One observation is that the datasets are "prepared" somewhat artificially (noise is programmatically inserted in the KGs, and the model is expected to detect these alterations), and it's not entirely clear how well this added noise correlates with the noise encountered in real-world KGs. "
>   Note that we do not introduce noise to NELL KG as it already comes with naturally occurring noise. We synthetically introduce noise to only FB15K-237, Yago3-10, and WN18RR datasets, by adopting and extending a previously proposed noise model from [Pujara et al., EMNLP 2017] (which only introduced random noise). Our noise model is designed in order to make it harder for all models to refine (by adding type-compatible noise) as described in Section 5.  While we agree that there is much work required in defining better models of noise on KG, and their impact assessment, it is beyond the scope of our current submission.
>
> 5. "However, I would have liked to see a clearer explanation of the algorithm and more implementation details. "
>
> We have now added Algorithm 1 in Appendix A.5 which should make the flow clearer.

---

### Official Review · AnonReviewer3 · 2020-04-01
**Using PSL to improve graph KG embeddings**

**Rating:** 6
**Confidence:** 4

**Review:**

The authors present a method to improve the performance of graph embeddings by using PSL to reason using ontology axioms to predict the types of entities. The Iterefin method is able to the predicted types as supervision to finetune the embeddings (ComplEx and ConvE). The authors propose an iterative method where the predictions from embeddings are fed back to the PSL model to infer new types.

The experiments are performed on corrupted versions of NELL, Yago3-10, FB15K-237 and wordnet using the methodology introduced in Pujara's KGI work. The experiments show substantial improvement on datasets with rich ontologies (not wordnet). The effects of iteration are minimal, so it is not clear that they are useful as some iterations result in slight improvements while others result in loss of performance.

The ablation studies show that range and subclass are the most important axioms, and other have minimal or no effect. Additional details would be useful about the number of each type of constraint in the PSL model as it is not clear whether the contribution is due to the number of character of the constraint. The importance of the range constraint seems correlated to the method for introducing noise in the evaluation datasets.

Pros:
- interesting approach for combining two different approaches to reasoning.
- good experiments to show the benefits of the method.

Cons:
- the claim that the iterative methods is helpful (which is part of the name of the system) is not supported by the experiments.
- no data on execution times and scalability (all experiments are on small or medium size datasets)
- insufficient analysis of the contribution of different axioms (table 5 is not enough).

The paper is well written and easy to follow. Substantial room is spent on the analysis of the iterative method, which in my opinion is not producing the desired results. The space could be used to describe the method in more detail and include additional experiment results.

---

> ### Author Response · Authors · 2020-04-08
> **Added analysis, and clarification on the claims.**
>
> We thank all the reviewers for their valuable comments on our submission. We have revised our submission taking into consideration some of the review comments (revised content is specifically marked in blue for easy identification).
>
> 1. "The effects of iteration are minimal, so it is not clear that they are useful as some iterations result in slight improvements while others result in loss of performance.
> - the claim that the iterative methods is helpful (which is part of the name of the system) is not supported by the experiments."
>
>    We agree that adding too many iterations is not always beneficial across all datasets. However, it is important to note that for certain datasets that have good ontology (viz., NELL and Yago3-10) the best performance of IterefinE is after completing at least 3 iterations (Figure 2). Further, considering only the negative facts (Table 3), IterefinE requires up to 3 iterations to achieve nearly 30% improvement over the second-best baseline for YAGO3-10 and over 100% for FB15K-237.  Referring to Figure 5, noise removal in IterefinE improves only at 3rd iteration for NELL. For KGs such as WN18RR with poor ontology, the iterations are indeed not helpful. Therefore, the iterative framework we have introduced is expected to be beneficial for refinement tasks over KGs with ontologies.
>
>
> 2. "The ablation studies show that range and subclass are the most important axioms, and other have minimal or no effect."
> "- insufficient analysis of the contribution of different axioms (table 5 is not enough)."
>
>   We disagree that DOM/RNG/SUBCLASS are the only important information (although it may have been the impression we inadvertently gave by the results in Table 5). In this revision, we have added another table, Table 8, which shows the performance using only one of the ontological constraints we have considered (and the combination of DOM and RNG).  As we can see from these results, using the only RNG is in fact no different than using only RMUT (on FB15K-237), or using any one of SAMEENT or MUT (on NELL). Thus, we argue that using all constraints from a rich ontology is important. We have added this discussion in the Appendix in the revised version.
>
> 3. "Additional details would be useful about the number of each type of constraint in the PSL model as it is not clear whether the contribution is due to the number of character of the constraint. "
>
>   Table 2 has details about the number of ontological constraints. In the paper, we have added an additional Table 8, which further looks at the impact of retaining only one (or a combination of DOM/RNG) rules. Further, we performed the following experiment on FB15K-237. We retained ALL rules except for the RNG rules. We then gradually added RNG rules to see how increasing the number of ontological constraints would impact the performance. We tabulate these results below. Taken in conjunction with Table 8, this clearly shows that a combination of all possible rules gives us the best result – particularly to remove noisy rules (-ve F1).
>
> % of RNG rules     -ve F1     +ve F1       wF1
> 0%                 0.72        0.97       0.95
> 25%                0.74        0.98       0.96
> 50%                0.75        0.98       0.96
> 75%                0.77        0.98       0.97
> 100%               0.80        0.98       0.97
>
>
> 4. "The importance of the range constraint seems correlated to the method for introducing noise in the evaluation datasets."
>
>    We introduce two kinds of noise: 1. “Random” noise: this is the noise that can sometimes be detected by checking the range constraints, and 2. “Type-compatible” noise: This kind of noise cannot be detected using range constraints – we specifically introduced this noise to make it difficult for the system to predict.
>
> Further, please note that for NELL, no noise is introduced since that KG already has naturally occurring noise as a result of the way in which it is extracted.
>
> 5. "- no data on execution times and scalability (all experiments are on small or medium-size datasets)"
>
>    We have now added execution times in Section 6. Briefly, the execution times depend on how efficient the embeddings methods are. For each iteration, TypeE-ComplEx takes between 25-100min, TypeE-ConvE takes between 120-480min.

---

### Author Response · Authors · 2020-06-09
**Link for supplementary material**

We have hosted the complete paper with Appendix on arxiv at https://arxiv.org/abs/2006.04509

---

### Decision · Program_Chairs · 2020-04-30

**Decision:**

Accept

**Comment:**

This paper proposes a novel method, IterefineE, for cleaning up noise in KGs. This method combines the advantages of using ontological information and inferences rules and KG embeddings with iterative co-training. IterefineE improves the task of denoising KGs on multiple datasets. While the importance of multiple iterations is mixed, reviewers agree that the combination of two significantly different types of reasoning is a promising direction.